# *In silico* characterization of bioactive phytochemicals as antivirals targeting the reovirus σ1 protein for inhibiting σ1-mediated host cell entry

Eitu Dey[1,2☯], Shipon Dey[1,2☯], Leu Nandi[1,2☯], Sayed Huzaifa Mumit[1,2], Refatul Arfat[1,2‡], Saifur Rahman Saif[1,2‡], Md. Monirul Islam[2,3*]

1 Department of Biochemistry and Biotechnology, University of Science and Technology Chittagong (USTC), Chattogram, Chittagong, Bangladesh, 2 Bangladesh Bioscience Research Group (BBRG), Chattogram, Chittagong, Bangladesh, 3 Department of Biochemistry and Molecular Biology, Noakhali Science and Technology University (NSTU), Noakhali, Chittagong, Bangladesh

☯ These authors contributed equally to this work.
‡ These authors also contributed equally to this work.
* monirul.bmb@nstu.edu.bd

## Abstract

*Mammalian orthoreoviruses* (MRVs), commonly known as reoviruses, are an emerging zoonotic threat that are known for their broad host tropism and potential for causing severe clinical pathology in both humans and animals. Despite this epidemic risk, currently, there are no approved therapeutic agents that are able to disrupt MRV transmission. The viral attachment protein sigma1 (σ1), mediating the entry of the virus into the host cells is a critical target for antiviral intervention. This study used an *in silico* structure-based drug design strategy to screen for bioactive phytochemicals that are capable of inhibiting the function of σ1. We screened a library of 376 bioactive phytochemicals with known antiviral potential against the σ1 receptor binding domain using molecular docking. Among the candidates, catechin gallate was the most potent inhibitor, possessing a superior binding affinity of −8.1 kcal/mol followed by bilobetin, which also showed a favorable binding affinity of −7.8 kcal/mol. Structural interaction analysis showed that catechin gallate and bilobetin occupies the active JAM-A binding pocket, forming stable interactions with some of the residues, including Gly381, Glu384, and Arg316, which are essential for the reovirus in the cellular attachment process. Subsequent pharmacokinetic and toxicity profiling proved that catechin gallate possessed favorable safety and drug-like characteristics, whereas bilobetin exhibited an unfavorable toxicity profile. In addition, molecular dynamics (MD) simulations supported the stability of σ1-catechin gallate complex relative to the σ1-bilobetin complex. Extensive post-trajectory analyses including RMSD, RMSF, Rg, SASA, and H-bond, showed that the binding of the catechin gallate significantly increases the rigidity and compactness of the protein. PCA indicated that the first three principal components (PC1-PC3) accounted for 74.1%

**Data availability statement:** All relevant data are within the paper and its Supporting Information files.

**Funding:** The author(s) received no specific funding for this work.

**Competing interests:** The authors have declared that no competing interests exist.

and 76.2% of the total variance for catechin gallate and bilobetin, respectively, with the σ1-catechin gallate complex displaying a more compact conformational cluster consistent with greater stability. MM-GBSA analysis also showed favorable binding for both complexes, with estimated binding energies of −15.6097 ± 3.21 kcal/mol and −13.7327 ± 5.44 kcal/mol for the σ1-catechin gallate and σ1-bilobetin complexes, respectively, with catechin gallate showing comparatively stronger binding. Our results reveal a precise mechanism by which the lead compound catechin gallate sterically occludes the σ1 receptor-binding pocket, thereby likely abrogating viral attachment to the host cell. This comprehensive preclinical evaluation provides supporting evidence for the further development of catechin gallate using *in vivo* models and clinical trials as a promising antiviral candidate against reovirus infection.

## Introduction

Viral pathogenesis is triggered by the specific binding of a virus to receptor molecules located on the surface of the host cell. This primary interaction is a key event that determines viral tropism and disease pathogenesis, and therefore, is a promising target for therapeutic intervention [1,2]. Despite the diversity of viral cell entry strategies, the most basic tenets are conserved, including the processes of target cell attachment, internalization, traversal of a lipid bilayer, and ultimately, delivery of the viral genome to a suitable intracellular site for replication [3,4]. Although viral attachment can be a simple, monophasic event, it is often accompanied by multiple receptors, a mechanism which often acts to enhance adhesion and promote entry [5]. For mammalian reovirus, this key process is mediated by the binding of the outer-capsid protein σ1 to junctional adhesion molecule A (JAM-A) [6].

The family *Reoviridae*, named after the acronym respiratory enteric orphan (reo) viruses, encompasses a broad range of pathogens infecting a variety of hosts, including humans, animals, plants, and insects [7]. This large family is a group of non-enveloped viruses with segmented double-strand RNA (dsRNA) genomes. Viruses belonging to the genus *Orthoreovirus*, which contains 10 genome segments, infect mammals, birds, and reptiles [8]. Members of this genus are further divided into two phenotypic types, fusogenic and non-fusogenic, based on their ability to cause cell-cell fusion and the production of large syncytia [9]. MRV are non-fusogenic as opposed to other members of the *orthoreovirus* family, including avian (ARV), baboon (BRV), reptilian (RRV), and bat orthoreoviruses, which are fusogenic [10]. MRV, which was first isolated from humans in the 1950s, is the primary model system for understanding the molecular biology of the family *Reoviridae* [11,12].

Reovirus particles are nonenveloped, icosahedral virions containing 10 dsRNA segments consisting of three large (L1, L2 and L3), three medium (M1, M2 and M3) and four small (S1, S2, S3, and S4) segments [13]. The structure of the virion is comprised of two concentric protein shells, an outer capsid and an inner core [14]. The outer layer is made up of heterohexamers of μ1 (encoded by M2) with σ3 (encoded by S4). At each fivefold axis, the attachment protein σ1 (encoded by S1)

projects from turret like assemblies formed by λ2 (encoded by L2) pentamers. The core shell is asymmetric λ1 (encoded by L3) dimers stabilized by σ2 (encoded by S2) with λ3 (encoded by L1) and μ2 (encoded by M1) attached to the inner core surface by λ1. Three serotypes circulate in humans and other mammals, T1, T2 and T3, as defined by neutralization and hemagglutination inhibition profiles [12]. The prototype strains are type 1 Lang (T1L), type 2 Jones (T2J), and type 3 Dearing (T3D), with total genome sizes of 23,606 bp, 23,578 bp, and 23,560 bp, respectively [7]. These serotypes vary greatly with respect to cellular tropism, cytopathic mechanisms, strategies for dissemination and propensity to cause central nervous system disease.

The viral attachment protein σ1 is an elongated filamentous molecule with a characteristic head and tail morphology [15,16]. Three structural regions of σ1 protein can be distinguished: a N terminal α helical coiled-coil tail, a central β spiral body, and a C-terminal globular head. Short flexible linkers join these domains and are postulated to allow for conformational flexibility required for receptor engagement [17]. Infection begins with the binding of the σ1 protein to receptors on the surface of the host cell [18]. In type 3 (T3) strains, σ1 recognizes two different receptors, α linked sialic acid (SA) [19] and JAM-A [20]. Residues in the β spiral body mediate SA binding, whereas determinants in the globular head interact with JAM-A [16,21]. JAM-A, the only receptor identified for reovirus to date, facilitates access of prototype and field isolates of all three serotypes [6]. A member of the immunoglobulin superfamily that participates in cell-cell adhesion, JAM-A is found in large amounts at epithelial and endothelial tight junctions that maintain barrier integrity [22], on hematopoietic cells where it is involved in leukocyte extravasation [23], and on platelets where it is involved in activation involved in thrombosis [24]. Structurally, JAM-A consists of a N-terminal ectodomain containing two immunoglobulin (Ig) like modules (membrane-distal D1 and membrane-proximal D2), a single pass transmembrane region, and a C terminal cytoplasmic tail containing a PDZ binding motif connecting to tight junction scaffolds [25,26]. JAM-A mediates cell adhesion by forming homotypic interactions between two JAM-A monomers on adjacent cells [27]. σ1 binds to the JAM-A D1 domain with an affinity about 1000-fold greater than JAM-A self-association, and this likely interferes with homodimers during virion attachment [21].

The multiple segmented structure of the MRV genome allows genetic reassortment and intragenic rearrangements. Such genomic plasticity leads to evolutionary dynamics which may engender unpredictable biological phenotypes and expansion of the host range [28,29]. Although the presence of rigid species barriers is not apparent, MRVs have been reported to infect humans as well as a broad spectrum of mammalian taxa including livestock, companion animals, and wildlife [30–32]. The oral route is by far the most common route of transmission; viral replication has been described in the intestinal epithelial cells and intranasal infection can produce severe influenza-like illness in susceptible hosts [33]. In murine models, certain MRV strains spread to the myocardium where they cause myocarditis, or to the central nervous system (CNS) where they infect neurons and cause lethal encephalitis in neonatal mice [34–37]. Infections have also been reported in a diverse range of species, including white-tailed deer [38], bats [11,39,40] and chamois [29]. MRVs are associated with respiratory, neurological, and enteric disease phenotypes in humans [32,41], but also in other mammalian species [30,42]. Direct and indirect contact, however, are the two modes of MRV zoonotic transmission. Due to the virion stability outside the host, MRVs are often isolated and detected in various environmental matrices, such as surface water, seawater, and wastewater [43–46], which helps them to remain persistent in the environment. Thus, active surveillance for emerging MRV in wildlife populations is very important to reduce spillover events into livestock and human populations. MRV type 3 (MRV3) is pathogenic in pigs, causing enteritis, pneumonia and encephalitis and has recently been linked to epidemic diarrheal and respiratory disease outbreaks in China, Korea and the United States [8,30,47,48]. MRV3 is potentially involved in outbreaks of acute gastroenteritis of piglets often co-infecting with porcine epidemic diarrhea virus (PEDV) and porcine deltacoronavirus (PDCoV). Furthermore, neurovirulent MRV3 strains have been isolated from dogs suffering from diarrhea in Japan [49], and Italy [50]. However, the full pathogenic potential of MRV3 is poorly understood and significant gaps in our knowledge still exist on its ecology, routes of transmission and possible wildlife reservoirs.

One significant hurdle in the development of antivirals is the ability of viruses to enter and develop evasion techniques [51]. The diversely characterized metabolites and compounds of the plants may be evaluated and utilized to overcome the evasion and drug-resistance issues linked with antivirals, thereby hindering the spread of the virus [52]. The antiviral activity of phytochemicals is mediated by numerous mechanisms [53]. Any synthetic medication derived solely from plant sources, aerial and non-aerial parts of plant, juices, resins and oil in its raw or pharmaceutical form is known as phytochemical therapy or herbal medicine [54]. Numerous phytochemicals have been generally shown in recent studies to offer some promise against a variety of virus targets, reducing the potential of resistance. When compared to synthesized medications, naturally derived compounds are often less toxic, improving their safety profiles. Plant-based chemicals are inexpensive and easily accessible, especially in conditions with low resources.

Currently, no approved antiviral medications or standardized treatments exists for mammalian reovirus infection in humans or other mammals. Although vaccines are used in poultry to control avian reoviruses associated with arthritis and stunting, no MRV vaccine is available [55,56]. Computer-aided drug design (CADD) presents a viable and rational strategy for developing effective therapeutic solutions against viral infection. However, to our knowledge, a critical gap exists in antiviral research, as no computational studies have yet evaluated phytochemical inhibitors specifically targeting the reovirus σ1 protein. To address this gap, we applied a suite of bioinformatic and chemoinformatic approaches to quantify the binding affinities and interaction profiles of drug like phytochemicals with σ1. The potential of lead compounds was further assessed through *in silico* pharmacokinetics and toxicity profiling [57]. In keeping with contemporary Computer Aided Drug Design (CADD), our workflow leverages virtual screening to triage large libraries, target identification, *in silico* toxicity assessment, and iterative optimization to accelerate lead discovery while reducing cost, timelines, and animal use [58]. We further tested the stability and plausibility of lead complexes under physiologic conditions using molecular docking and molecular dynamics (MD) simulations, analyzing RMSD, RMSF, hydrogen bond occupancy, solvent accessible surface area (SASA), radius of gyration (Rg), principal component analysis (PCA), and binding free energy (MM-GBSA).

This research aims to find natural compounds that have the ability to interfere with the binding and entry of viruses and contribute to the development of broad-spectrum anti-viral compounds with greater precision against reoviruses. It is very important to note however, that computational techniques such as molecular docking and MD simulations have limitations. These include simplified scoring functions, imperfect treatment of solvation effects and a high sensitivity on initial conformations. Therefore, *in silico* predictions need to be strictly validated with experimental assays in order to validate antiviral potency, pharmacokinetic properties and safety in vitro and in vivo conditions. Such experimental corroboration is critical for translating these computational findings into clinically-relevant therapeutics.

## Materials and methods

A schematic representation of the study workflow is shown in Fig 1.

### Extraction and purification of Protein structure

Attachment Protein σ1 homotrimer (PDB ID: 1KKE; Method: X-ray diffraction; Resolution: 2.60 Å; Organism: Mammalian orthoreovirus 3 Dearing) was obtained from the RCSB Protein Data Bank (https://www.rcsb.org/) [59,60]. This homotrimer contains three identical chains of 210 amino acids each [15]. Protein preparation (Fig 2) was carried out by removing water molecules, side chains, and heteroatoms using PyMOL [61] and Discovery Studio software [62]. Subsequently, the monomeric σ1 structure was optimized by correction of hydrogen bonding, side-chain conformations, and bond orders, followed by energy minimization using the GROMOS 43B1 force field in Swiss-PDB Viewer [63].

### Retrieval and preparation of ligand

A library of bioactive phytochemicals with known anti-viral activity was compiled using a systematic literature review. After exclusion of duplicates from the initial screening, a final data set of 373 unique compounds was established (S1 File). The

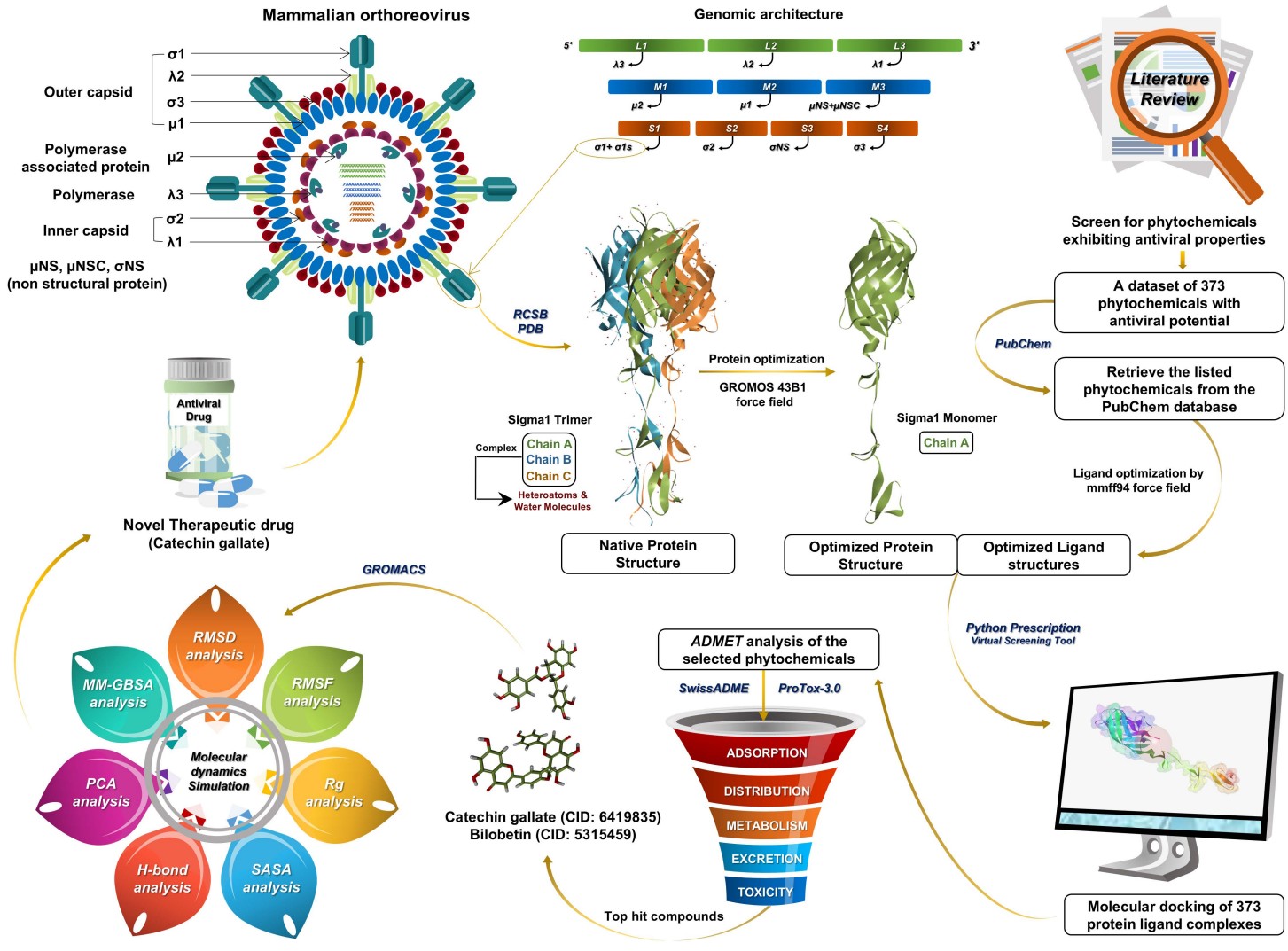

**Fig 1. Schematic workflow used to identify potential phytochemical inhibitors of the reovirus cell-attachment proteinσ1.**

corresponding three-dimensional structures were retrieved from the PubChem database (https://pubchem.ncbi.nlm.nih.gov/) [64]. Prior to docking the ligand geometries were optimized and energy-minimized using the PyRx platform utilizing the MMFF94 force field [65]. The minimization protocol used a steepest descent algorithm involving a total of 2,000 minimization steps to achieve stable conformations.

## Prediction of the pharmacokinetics (Pk) parameters

The pharmacokinetic properties of the lead compounds were assessed using the SwissADME web server (http://www.swissadme.ch/) to predict absorption, distribution, metabolism, and excretion (ADME) profiles [66]. Canonical SMILES strings were used as input data for assessing Physicochemical Properties, Lipophilicity, Water Solubility and important Pharmacokinetic parameters. In addition, compliance with Lipinski's Rule of Five was also evaluated to estimate oral bioavailability and therapeutic viability [67].

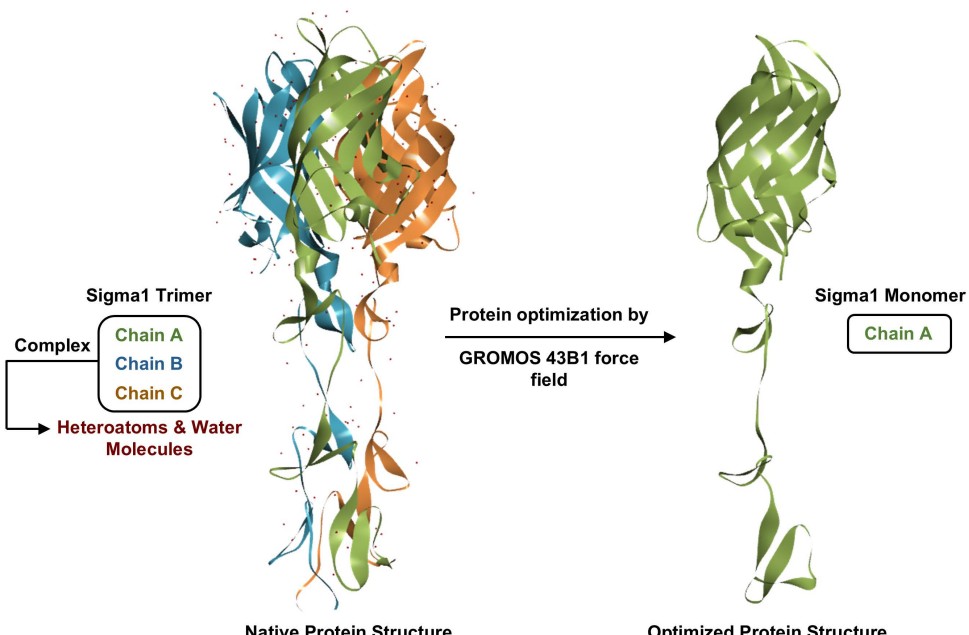

**Fig 2. Preparation of the σ1 monomer from the native trimeric structure via removal of heteroatoms and water molecules, GROMOS 43B1 optimization, and isolation of chain A for molecular docking.**

## Prediction of toxicological properties

The safety profiles of the selected candidates were analyzed using the ProTox-3.0 (https://tox.charite.de/protox3/) computational platform [68]. This step was important in order to optimize the safety and therapeutic potential of the lead compounds. This server offered predictive modeling of acute oral toxicity ($LD_{50}$) and toxicity classification. The assessment included a wide range of organ-specific toxicities (hepatotoxicity, cardiotoxicity, nephrotoxicity and neurotoxicity) and Toxicity end points (mutagenicity, carcinogenicity, cytotoxicity and immunotoxicity) to provide a comprehensive assessment of the biological safety.

## Molecular docking study

Molecular docking was used to predict the binding modes and affinities of ligands at the potential active site of the σ1 protein. Using the PyRx Virtual Screening Tool [69] the parameters were set to dock flexible ligands with a rigid protein structure. The search space was defined with a grid box with coordinates X: 20.942, Y: 13.948, and Z: 2.418, with dimensions of 28.1885 Å × 26.7418 Å × 20.2474 Å. The grid box was generated to cover all the key residue of the σ1 protein active site involved in binding to junctional adhesion molecule A (JAM-A) during cell attachment [70]. Before docking the ligands were energy minimized and converted to PDBQT format. The resulting protein-ligand complexes were visualized using BIOVIA Discovery Studio with the lowest binding free energy (kcal/mol) being selected as the best conformation.

## Molecular dynamics simulation

All-atom molecular dynamics (MD) simulations were carried out using GROMACS 2024.4 in order to determine the stability of the optimal protein-ligand complex [71]. The protein topology was generated using the CHARMM36 force field, while ligand parameters were derived via CGenFF to ensure full compatibility [72,73]. The system was solvated in a triclinic TIP3P box with ≥1.0 nm padding, neutralized with $Na^+/Cl^-$, and adjusted to 0.15 M NaCl. Following steepest-descent

minimization, systems underwent 100 ps NVT and 100 ps NPT equilibration. Production runs were 200 ns with a 2 fs timestep, writing coordinates every 10 ps. During post-processing, standard GROMACS tools were utilized for the analysis of RMSD, RMSF, radius of gyration (Rg), SASA, and hydrogen bond (H-bond). Global motions were examined via backbone PCA to capture collective dynamics and sampled conformations. Binding free energies were calculated by using gmx_MMPBSA 1.6.3, based on representative snapshots extracted across the production trajectory [74]. Together, these dynamical, energetic, and statistical assessments provide a unified view of binding stability, interaction energetics, and ligand-driven effects on protein conformational behavior.

## Results

### Molecular docking analysis

Virtual screening of 376 compound library found Catechin gallate to be the best candidate with a superior binding affinity of −8.1 kcal/mol. It was followed by Bilobetin with a docking score of −7.8 kcal/mol demonstrating a strong interaction potential making it the second most promising candidate (Table 1). Post docking structural analysis using BIOVIA Discovery Studio showed that both ligands establish a stable and extensive interaction network within the σ1 active site. Catechin gallate formed a very strong network of non-covalent contacts comprising four conventional hydrogen bonds (Asp346, Gly381, Glu384 and Thr455), two π-π T-shaped involving Tyr298, a π-π stacked interaction involving Phe454, as well as dual π-cation interactions involving Arg316 and Arg452 (Fig 3B). Comparatively, bilobetin showed a similar strong interaction profile, with five conventional hydrogen bonds with Asn312, Thr455, and Glu384, and two π-cation interactions with Arg316, a π-anion interaction with Asp382, and other π-based interactions (π-sigma, π-π stacked, and π-π T-shaped) Tyr298 (Fig 3D). Notably, the two ligands seem to competently occupy and block the binding pocket by

**Table 1. Identification of lead compounds (catechin gallate and bilobetin) based on name, PubChem CID, binding affinity, and their nonbonded interactions with the reovirus attachment protein σ1.**

| Compound name | Compound CID | Binding Affinity (Kcal/mol) | Residues in Contact | Distance in Å | Interaction Type |
|---|---|---|---|---|---|
| Catechin gallate | 6419835 | −8.1 | THR455 | 2.29743 | Conventional Hydrogen Bond |
| | | | GLY381 | 2.74805 | Conventional Hydrogen Bond |
| | | | GLU384 | 2.36837 | Conventional Hydrogen Bond |
| | | | ASP346 | 2.59142 | Conventional Hydrogen Bond |
| | | | ARG316 | 3.76126 | Pi-Cation |
| | | | ARG452 | 4.60186 | Pi-Cation |
| | | | PHE454 | 5.69355 | Pi-Pi Stacked |
| | | | TYR298 | 5.94866 | Pi-Pi T-shaped |
| | | | TYR298 | 4.82723 | Pi-Pi T-shaped |
| Bilobetin | 5315459 | −7.8 | ASN312 | 2.59512 | Conventional Hydrogen Bond |
| | | | THR455 | 2.59631 | Conventional Hydrogen Bond |
| | | | THR455 | 2.23053 | Conventional Hydrogen Bond |
| | | | GLU384 | 2.51452 | Conventional Hydrogen Bond |
| | | | THR455 | 2.37206 | Conventional Hydrogen Bond |
| | | | ARG316 | 3.55796 | Pi-Cation |
| | | | ARG316 | 4.08716 | Pi-Cation |
| | | | ASP382 | 4.51933 | Pi-Anion |
| | | | TYR298 | 3.69831 | Pi-Sigma |
| | | | TYR298 | 4.38528 | Pi-Pi Stacked |
| | | | TYR298 | 5.06066 | Pi-Pi T-shaped |

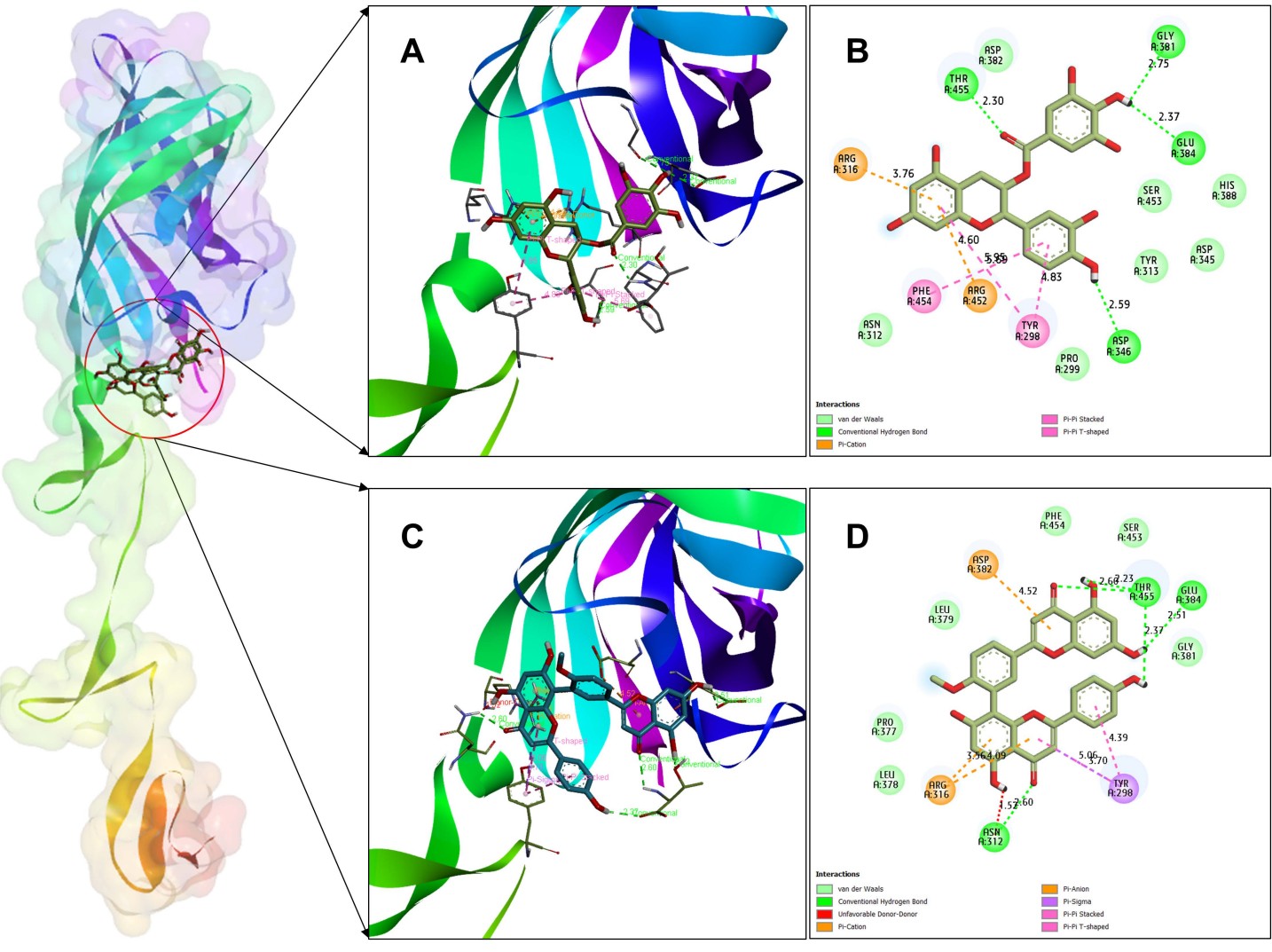

**Fig 3. Three-dimensional and two-dimensional interactions between the reovirus attachment protein σ1 and the lead compounds catechin gallate and bilobetin at the JAM-A binding domain. (A, B) catechin gallate; (C, D) bilobetin.**

interacting with key residues, including Gly381, Glu384, and Arg316, that are involved in reovirus host cell attachment [70]. Based on their favorable binding energetics and strategic positioning within the receptor active site, the σ1-catechin gallate and σ1-bilobetin complexes were selected for subsequent molecular dynamics simulations to further evaluate their conformational stability.

## Assessment of pharmacokinetics (Pk) properties

Pharmacokinetic profiles of candidate ligands were extensively analyzed to assess drug likeness of the candidates based on absorption, distribution, metabolism, and excretion (ADME) characteristics. SwissADME was used to predict physicochemical properties, lipophilicity, solubility, pharmacokinetics, medicinal chemistry friendliness, and general drug-like properties. The resulting data (Table 2) indicate that both catechin gallate and bilobetin exhibit favorable drug-likeness properties, satisfying key criteria associated with bioavailability and supporting their potential for pharmaceutical development.

**Table 2. Pharmacokinetic and toxicity properties of the lead compounds catechin gallate and bilobetin.**

| Properties | | Catechin gallate | | Bilobetin | |
|---|---|---|---|---|---|
| **Physicochemical Properties** | Formula | $C_{22}H_{18}O_{10}$ | | $C_{31}H_{20}O_{10}$ | |
| | MW | 442.37 | | 552.48 | |
| | Heavy atoms | 32 | | 41 | |
| | Aromatic heavy atoms | 18 | | 32 | |
| | Rotatable bonds | 4 | | 4 | |
| | H-bond acceptors | 10 | | 10 | |
| | H-bond donors | 7 | | 5 | |
| | MR | 110.04 | | 151.44 | |
| | TPSA | 177.14 | | 170.8 | |
| **Lipophilicity** | Consensus Log P | 1.25 | | 3.96 | |
| **Water Solubility** | ESOL Log S | −3.7 | | −6.96 | |
| **Pharmacokinetics** | GI absorption | Low | | Low | |
| | BBB permeant | No | | No | |
| | Pgp substrate | No | | No | |
| | log Kp (cm/s) | −7.91 | | −5.86 | |
| **Druglikeness** | Lipinski violations | Yes;1 violation | | Yes; 1 violation | |
| | Bioavailability Score | 0.55 | | 0.55 | |
| **Medicinal Chemistry** | Synthetic Accessibility | 4.16 | | 4.35 | |
| **Oral toxicity prediction** | $LD_{50}$ | 1000 mg/kg | | 4000 mg/kg | |
| | Toxicity Class | 4 | | 5 | |
| | | Prediction | Probability | Prediction | Probability |
| **Organ toxicity** | Hepatotoxicity | Inactive | 0.7 | Inactive | 0.8 |
| | Neurotoxicity | Inactive | 0.88 | Inactive | 0.85 |
| | Nephrotoxicity | Active | 0.73 | Active | 0.62 |
| | Cardiotoxicity | Inactive | 0.89 | Active | 0.59 |
| **Toxicity end points** | Carcinogenicity | Inactive | 0.54 | Inactive | 0.65 |
| | Immunotoxicity | Inactive | 0.87 | Active | 0.84 |
| | Mutagenicity | Inactive | 0.7 | Inactive | 0.81 |
| | Cytotoxicity | Inactive | 0.82 | Inactive | 0.93 |

## Assessment of toxicological properties

*In silico* toxicity predictions were performed by using the ProTox-3.0 platform to determine acute oral toxicity, organ-specific toxicities and general toxicity endpoints (Table 2). Catechin gallate was listed under toxicity class 4 with an oral $LD_{50}$ of 1000 mg/kg and this profile indicates a relatively safe oral administration profile. The compound was predicted to be inactive to most standard toxicity parameters, with the specific exception of potential nephrotoxicity. This adverse effect requires careful consideration and possibly can be overcome by future structural modifications. Conversely, bilobetin was anticipated to have several toxicity liabilities, such as nephrotoxicity, cardiotoxicity and immunotoxicity, despite an oral $LD_{50}$ of 4000 mg/kg. Taken together, these results indicate that catechin gallate has a more acceptable safety profile than bilobetin, justifying its further development as an antiviral lead scaffold.

## Molecular dynamics simulation analysis

In order to understand the dynamic stability and conformational behavior of the reovirus σ1 protein upon binding to a ligand, a 200 ns molecular dynamics (MD) simulation was performed. The resulting trajectories were quantitatively

evaluated by RMSD, RMSF, Rg, SASA, hydrogen bonding profiles, PCA, and MM-GBSA in order to characterize the inter-molecular interactions and structural changes induced by catechin gallate and bilobetin.

## Root mean square deviation (RMSD) analysis

The conformational behavior of the unligated (apo) protein and its ligand-bound complexes was evaluated by monitoring Cα backbone RMSD throughout the simulation (Fig 4A). The apo protein σ1 showed a high conformational lability, with a rapid initial deviation to a maximum of 1.611 nm in the first 20 ns. This was followed by high amplitude oscillation (0.8–1.2 nm) and an average RMSD value of 0.949 ± 0.166 nm, indicating inherent structural flexibility in the absence of the ligand. Conversely, the σ1-catechin gallate complex reached equilibrium in a short time (20−30 nanoseconds) and maintained a stable trajectory during the rest of the simulation. The complex exhibited a smaller fluctuation range (0.6–0.8 nm) and also a considerably lower average RMSD of 0.680 ± 0.178 nm. In comparison, the σ1-bilobetin complex showed significantly higher RMSD values during the 200 ns simulation with an average RMSD of 4.106 ± 0.208 nm. The attenuation of RMSD amplitude in the σ1-catechin gallate complex is suggestive of the Catechin gallate binding has structural constraints that minimize large scale conformational deviations.

## Root mean square fluctuation (RMSF) analysis

RMSF provides residue level flexibility analysis through quantification of average deviation of Cα atoms positions over the simulation period. (Fig 4B). The apo protein showed high local fluctuations, resulting in an average RMSF of 0.62 ± 0.297 nm. In contrast, the σ1-catechin gallate complex showed a significant decrease of atomic mobility where the average RMSF decreased to 0.30 nm. Specific segments, particularly residues 310–386 and 295–452 showed minimal displacement, indicating that they are highly stable secondary structural elements in the σ1-catechin gallate complex. This global dampening of fluctuations of residue confirms the fact that the ligand provides rigidity at the binding pocket and stabilizes the overall protein architecture. In contrast, the σ1-bilobetin complex was relatively more flexible, with an average RMSF of 0.57 ± 0.345 nm, suggesting a less stabilizing influence and a more dynamic behavior similar to the apo protein.

## The radius of gyration (Rg) analysis

Structural compactness and folding dynamics were determined by analyzing the Radius of Gyration (Rg) (Fig 4C). Rg measures the compactness of the protein as the mass-weighted root mean square distance from the center of mass. The apo protein exhibited early-phase instability before it reached an average Rg value of 2.07 ± 0.19 nm. Distinctively, the σ1-catechin gallate complex was compressed very quickly in the first 20 ns with a subsequent equilibrium structure at a reduced average Rg of 1.89 ± 0.20 nm. Conversely, the σ1-bilobetin complex exhibited consistent fluctuations throughout the 200 ns trajectory and had a higher average Rg of 2.14 ± 0.22 nm, compared to the apo protein. The invariantly decreasing Rg values and slight Rg fluctuations in the σ1-catechin gallate complex point to ligand binding leading to a tighter molecular packing which inhibits the structural expansion characteristic of the unbound state.

## Solvent accessible surface area (SASA) analysis

The exposure of the protein surface to the aqueous environment was determined by SASA analysis (Fig 4D). Changes in values of SASA are indicative of changes in protein compactness, hydrophobic exposure and ligand-induced conformational adaptation. Both complexes maintained a constant solvation profile over the 200 ns trajectory. The σ1-catechin gallate complex (114.44 ± 1.4 nm$^2$) showed a marginal increase in mean SASA value compared to the σ1-bilobetin complex (114.33 ± 1.3 nm$^2$), whereas the apo form had a relatively lower average value of 112.89 ± 1.41 nm$^2$. While this slight increase suggests minor localized conformational rearrangements that would expose hydrophobic patches, the overall stability of the SASA plot confirms that the association of the ligands did not induce a significant solvent induced destabilization or unfolding.

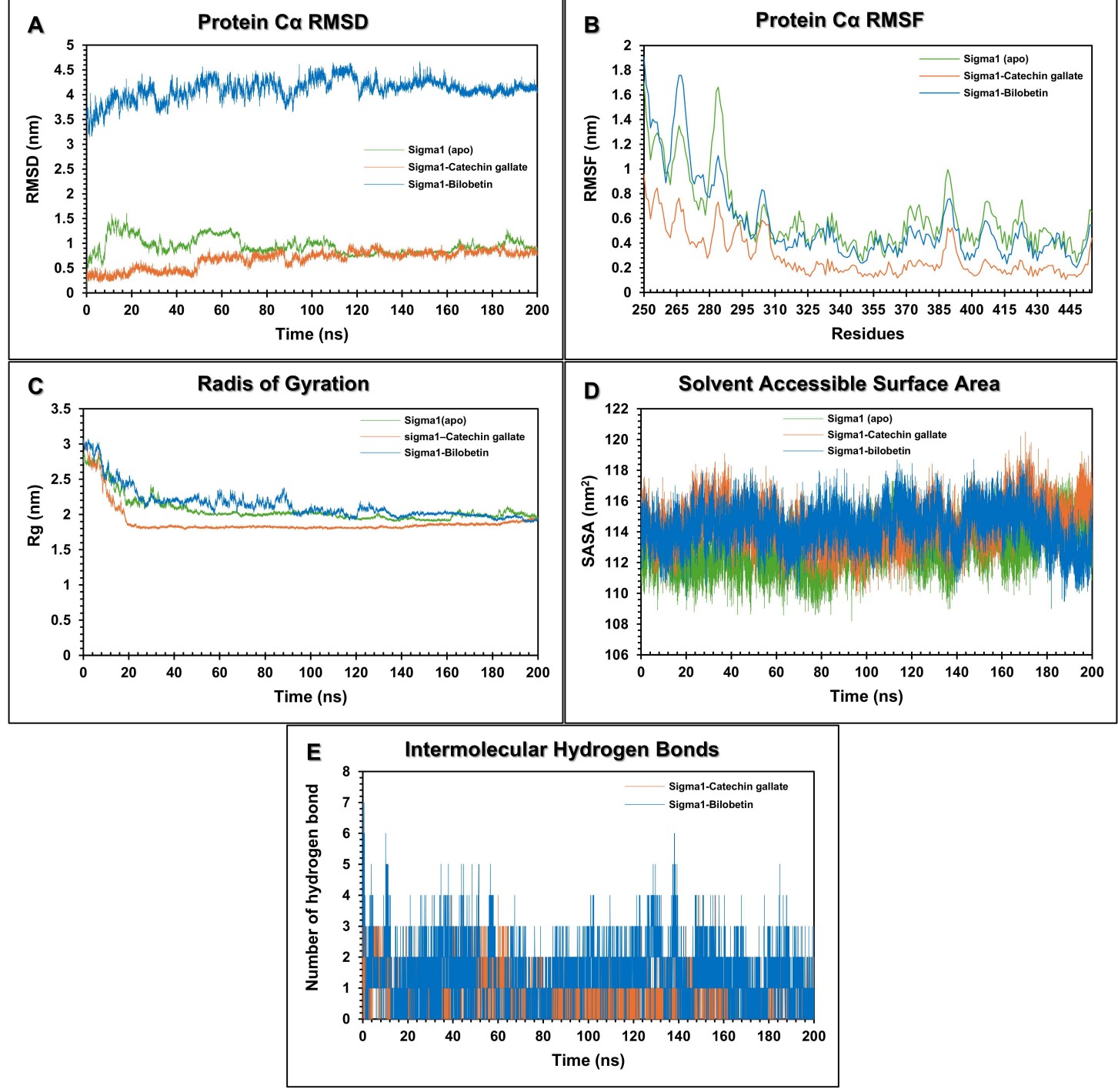

**Fig 4.  Molecular dynamics trajectory analyses of the apo σ1 protein, σ1–catechin gallate complex, and σ1–bilobetin complex. (A) RMSD, (B) RMSF, (C) radius of gyration (Rg), (D) solvent-accessible surface area (SASA), and (E) intermolecular hydrogen bonds.**

## Hydrogen bond analysis

The stability of the protein-ligand complex was further analyzed by monitoring the evolution of hydrogen bond interactions (Fig 4E). Hydrogen bonds are crucial non-covalent interactions that are responsible for maintaining biomolecular complex stability and specificity. The analysis showed the persistence of critical non-covalent contacts throughout the simulation for both σ1-catechin gallate and σ1-bilobetin complexes. An average of 0.73 hydrogen bonds was maintained between catechin gallate and the σ1 protein, indicating sustained interaction throughout the simulation. Significant peaks in the occurrence of hydrogen bonds were detected during the time interval 0–10 ns, 40–60 ns and 150 ns, corresponding to phases of active stabilization. The σ1-bilobetin complex also sustained hydrogen-bond interaction on the simulation trajectory, having an average of 1.43 hydrogen bonds. The persistence of such interactions suggests that both ligands adopt a thermodynamically favorable active-site orientation and is responsible for the excellent integrity of the complexes.

## Principle component analysis (PCA)

Dominant collective motions were deconvoluted with PCA in order to map essential dynamics of the system. PCA was applied to reduce the dimensionality of the MD trajectory and isolate the large-scale atomic fluctuations. By diagonalizing the covariance matrix, we obtained eigenvectors for the dominant modes of motion and eigenvalues for the structural variance associated with each mode. Scree plot analysis was then used to quantify the contribution of the top-ranking principal components (PCs) to the essential dynamics of the system. Both the apo and σ1-catechin gallate complex trajectories showed that the first three principal components (PC1, PC2, and PC3) accounted for approximately 74% of the total variance (Apo: 74.9%; σ1-catechin gallate complex: 74.1%), suggesting that the motion of the protein is characterized by a few low frequency modes. A similar trend was observed in the σ1-bilobetin complex, where the first three principal components explained 76.2% of the total variance, indicating a similarly significant contribution of a limited number of dominant motions. However, 2D representations of the trajectory into the PC1/PC2 phase space showed different conformational behaviors (Fig 5). The apo protein showed a diffuse and scattered distribution, indicating a high entropy state of a large conformational state space. In sharp contrast, the σ1-catechin gallate complex was found to be in a highly restricted and dense conformational cluster. This shift indicates that the available phase space is drastically reduced by the binding of the ligand and the protein is thus locked into a stable and well-defined conformation. Similarly, projection of the σ1-bilobetin trajectory onto the PC1/PC2 plane showed a widely scattered and irregular distribution, resembling the apo protein and signifying retention of a relatively flexible conformational landscape with an expanded accessible phase space (Fig 5C).

## Free energy estimation by MM/GBSA analysis

Molecular Mechanics-Generalized Born Surface Area (MM-GBSA) is a well-established technique for the determination of protein-ligand binding energies [75]. The thermodynamic stability of the complex was quantified using the MM-GBSA approach based on 200 snapshots extracted from the stable production trajectory and according to the equation:

$$\Delta G_{bind} = \Delta E_{MM} + \Delta G_{solv}$$

Where $\Delta E_{MM}$ is the energy of gas-phase molecular mechanics (GGAS) and $\Delta G_{solv}$ is the solvation free energy (GSOLV). ((Fig 6); labels VDWAALS, EEL, EGB, and ESURF correspond to $\Delta E_{VDW}$, $\Delta E_{EEL}$, $\Delta G_{GB}$, and $\Delta G_{SA}$, respectively). The analysis resulted into a strongly favorable binding affinity for σ1-catechin gallate complex with an estimated binding energy ($\Delta G_{bind}$) of −15.6097 ± 3.21 kcal/mol. A similar favorable binding was also observed in the σ1-bilobetin complex, but with a

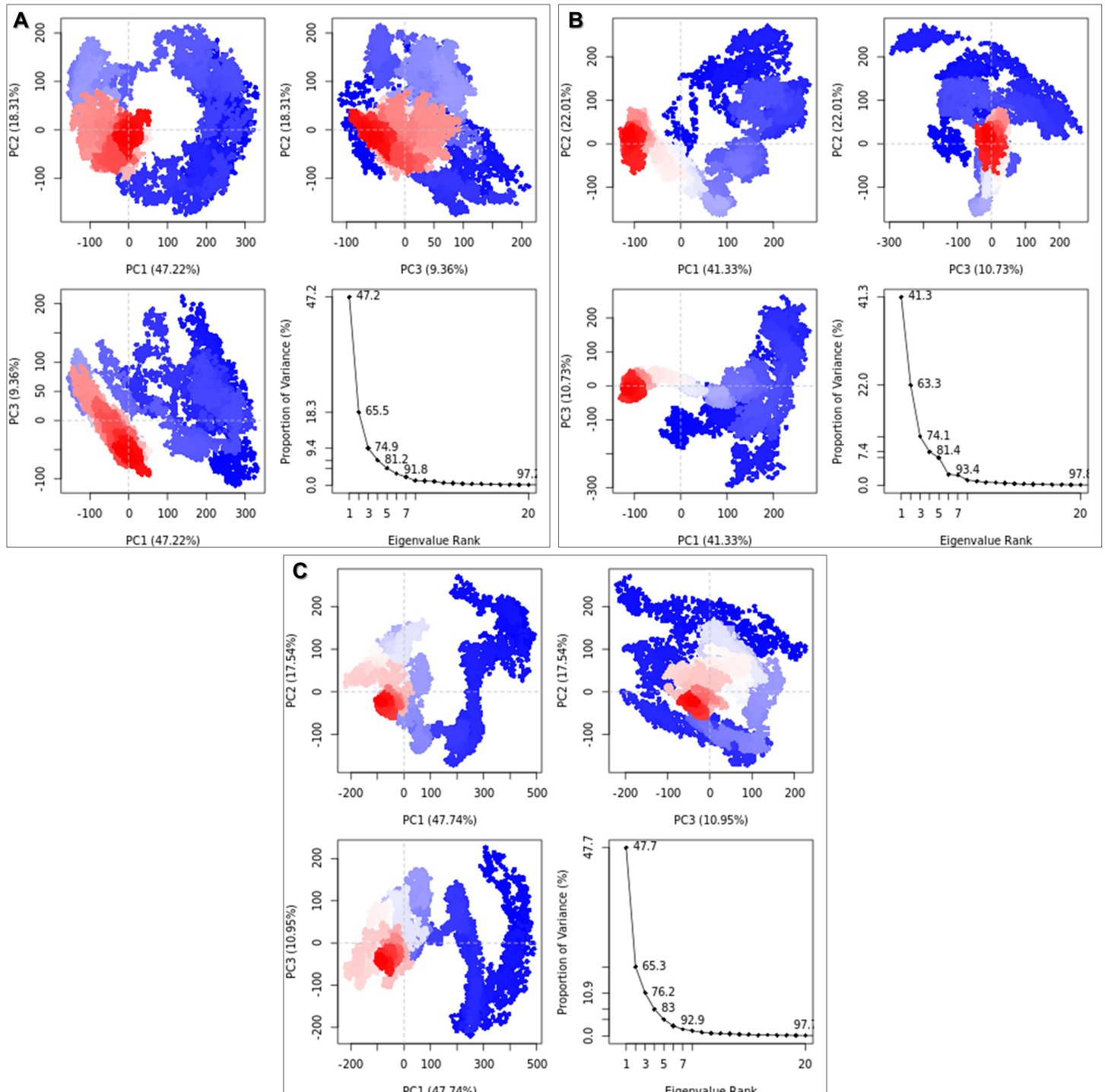

**Fig 5. Scree plot of principal component analysis (PCA).** Eigenvalues plotted against the proportion of variance explained by each principal component for (A) the apo form of σ1, (B) the σ1-catechin gallate complex, and (C) the σ1-bilobetin complex.

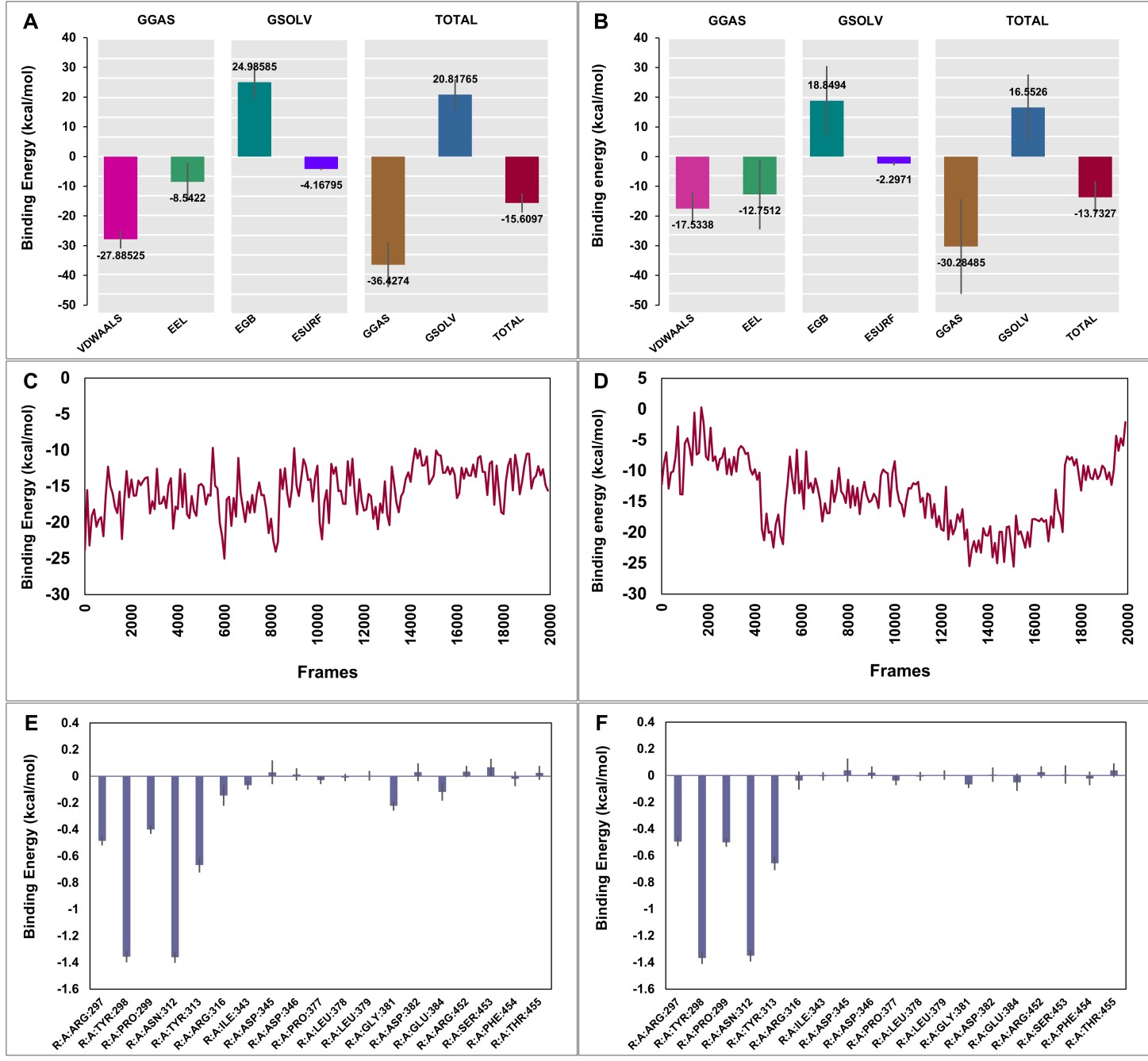

**Fig 6. MM/GBSA analysis after molecular dynamics simulations.** (A) Contributions of van der Waals, electrostatic, and solvation energies to the MM/GBSA binding free energy of the σ1-catechin gallate complex; (B) corresponding energy contributions for the σ1-bilobetin complex; (C) time-dependent variation in the total binding free energy of the σ1-catechin gallate complex; (D) corresponding variation for the σ1-bilobetin complex; (E) per-residue contributions of key amino acids to the binding free energy of the σ1-catechin gallate complex; and (F) corresponding per-residue contributions for the σ1-bilobetin complex. Error bars indicate standard deviations.

relatively lower binding affinity, giving an estimated $\Delta G_{bind}$ value of $-13.7327 \pm 5.44$ kcal/mol. In both complexes, energy decomposition showed that binding is attributed mainly to favorable gas phase interactions ($\Delta E_{MM}$) with significant contribution from both van der Waals ($\Delta E_{VDW}$) and electrostatic forces ($\Delta E_{EEL}$). The solvation term ($\Delta G_{solv}$) opposed complex formation, a phenomenon which was mainly due to the large desolvation penalty of the polar groups ($\Delta G_{GB}$). However, this penalty was partially compensated by a favorable non-polar solvation component ($\Delta G_{SA}$) which is associated with the burial of hydrophobic surface area upon ligand binding (Fig 6A and 6B). This energetic profile is a validation of the fact that the σ1-catechin gallate interaction is stabilized mostly by complementary electrostatic and hydrophobic forces within the binding pocket.

The time-resolved binding energy analysis also demonstrated the presence of different stability profiles in both systems. The σ1-catechin gallate complex always retained negative binding energies throughout all sampled frames with little variation, showing a stable interaction over the course of the simulation (Fig 6C). Conversely, the σ1-bilobetin complex was more diverse, indicating multiple frames with nearly neutral binding energies, indicating a weaker interaction pattern (Fig 6D). The energy decomposition in the residues revealed major contributions of active site residues, especially Tyr298 and Asn312, which had the most favorable energetic contributions in both complexes (Fig 6E and 6F).

## Discussion

Mammalian reoviruses are ubiquitous opportunistic pathogens with a wide range of hosts in animal species [76,77]. Although MRVs are usually linked with mild respiratory and enteric infections, they pose an emerging health concern to the population. Previous study has implicated reovirus infection in the pathogenesis of chronic human diseases, including Celiac disease [78], and are known to cause severe pathogenic effects, including pneumonia, myocarditis and encephalitis in young human hosts [34–37,79]. Moreover, the importance of wild animal populations as the MRV reservoirs requires a proper understanding of viral circulation to reduce the risk of zoonotic or livestock transmission.

Although the virus is widespread and has the potential to cause an epidemic, currently, there are no effective therapeutic drugs to prevent or treat reovirus infection. The development of new antiviral agents is of great importance in preventing the spread of these viruses and preventing related effects. Prevention of the initial attachment and entry of the virion into the host cell is one of the most effective ways of interrupting viral replication cycles [2]. The reovirus σ1 protein, the main viral attaching protein is an optimal therapeutic target as it mediates viral entry by interacting with the JAM-A. In this study, we screen and profile a library of bioactive phytochemicals to test their binding capacity to the JAM-A binding domain of the σ1 protein. This study aims to determine compounds that can competitively inhibit this protein-host receptor interaction and thereby block cellular entry via σ1 to provide a basis of new antiviral development.

The combination of computational prediction and experimental validation has completely transformed modern drug discovery, providing a fast and cost-effective method in the place of traditional high-throughput screening. Key to this approach is molecular docking, which clarifies the conformations of the ligands in the macromolecular binding pockets and measures intermolecular recognition to estimate binding free energies [80]. In this process, ADMET profiling helps in the early elimination of compounds having poor pharmacokinetic and toxicological trajectories. In order to further refine such predictions, molecular dynamics (MD) simulations are used to take into consideration the conformational flexibility of a protein, optimize the docked complexes and rigorously rank candidates according to binding stability before conducting wet-lab testing [81]. Despite having these advantages, *in silico* methodologies possess few limitations in drug discovery. A major bottleneck in approaches like inverse docking is the reliance on extensive and preexisting databases of three-dimensional structures of protein, or binding motifs [82]. Furthermore, the proper

modelling of solvation effects and thermodynamic properties requires high fidelity input structures and imposes a huge computational burden. MD simulations in particular are computationally expensive and are very sensitive to initial system parameters. Additionally, the heterogeneous nature of the data sources makes integration more difficult and relying too much on computational predictions without proper experimental support can lead to misinterpretation of results [83].

Structure-based molecular docking is a fundamental paradigm in rational drug design for elucidating binding dynamics and affinity of ligand-protein complex [84]. Utilizing this approach, we performed a virtual screen of 373 phytochemicals with known anti-viral properties to determine their interaction with the mammalian reovirus σ1 attachment protein. As described in Table 1, Catechin gallate (PubChem CID: 6419835) was the most favorable candidate and had the best thermodynamic profile with a binding energy of −8.1 kcal/mol. This indicates that Catechin gallate possesses a very strong and high-affinity binding to the viral target. Subsequent interaction profiling showed that Catechin gallate preferentially binds in the σ1 receptor-binding domain interacting with specific residues which are crucial for viral adhesion. These results indicate a mechanism in which the compound sterically occludes a viral interface therefore effectively inhibiting host cell entry. Bilobetin, on the other hand, had a binding affinity of −7.8 kcal/mol with the σ1 protein and established stable interactions with residues in the active site. These results imply that bilobetin can also fit in the receptor-binding pocket, thus likely disrupting reovirus attachment and subsequent host cell entry.

The evaluation of pharmacokinetic (PK) profiles represents a pivotal stage in the drug discovery process to ensure that candidate molecules have sufficient absorption, distribution, metabolism, and excretion (ADME) properties to advance through the clinically driven development process [85]. To obtain an estimate of the oral bioavailability of the target phytochemical, we applied Lipinski's Rule of Five (Ro5). This widely accepted heuristic suggests an ideal orally active drug should have a molecular weight (MW) < 500 g/mol, lipophilicity (consensus logP) < 5, < 5 hydrogen bond donors (HBD) and <10 hydrogen bond acceptors (HBA) [67]. In our study, the *in silico* analysis of catechin gallate showed a MW of 442.37 g/mol with a consensus logP value of 1.25. The hydrogen bonding parameters were calculated at 7 donors and 10 acceptors, respectively. Though there are slight variations in HBD counts, the compound generally matches the physico-chemical needs for good accessibility based on Lipinski rule. Furthermore, the partition coefficient, which is dependent on the equilibrium between n-octanol and water, and the aqueous solubility underlie a molecule's capacity to cross the lipid bilayers of cellular membranes [66]. Regarding these parameters, catechin gallate showed a Log S water solubility value of −3.7. Subsequent ADME profiling revealed that the compound is not a substrate for P-glycoprotein (P-gp) and has no blood-brain barrier (BBB) permeability and therefore, a low risk of toxicity in the central nervous system. Although predicted gastrointestinal absorption was classified low, the compounds showed an oral bioavailability score of 0.55 and a synthetic accessibility score of 4.16 and 4.35 for catechin gallate and bilobetin, respectively. Collectively, this *in silico* ADME analysis suggests that catechin gallate and bilobetin possess viable pharmacokinetic properties that support further development as therapeutic candidates against reovirus.

Toxicological profile assessment by *in silico* approaches has now become an integral part of the contemporary lead optimization and regulatory studies, providing a practical substitute for the ethical constrains and resource-intensive nature of traditional *in vivo* and *in vitro* screening. In this research, we used the ProTox-3.0 platform to obtain a comprehensive toxicity profile of the candidate phytochemicals. The compound had an overall favorable organ-safety profile with probability scores of 0.70, 0.88, and 0.89 of inactivity in hepatotoxicity, neurotoxicity and cardiotoxicity, respectively. Although this *in silico* evaluation indicated a possible liability of nephrotoxicity (probability 0.73) indicating a precise target of structural optimization, catechin gallate showed promising outcomes across multiple toxicity endpoints. In particular, the analysis anticipated inactivity in carcinogenicity (0.54), immunotoxicity (0.87), and mutagenicity (0.70), as well as absence of cytotoxicity (0.82). Conversely, Bilobetin exhibited predicted toxic liabilities, such as possible neurotoxicity and cardiotoxicity with probability scores of 0.62 and 0.59, respectively. It further showed significant risk of immunotoxicity

(probability 0.84), indicating relatively poor safety properties compared to catechin gallate. Collectively, this *in silico* analysis indicates that catechin gallate has a promising predicted safety profile, which supports its potential suitability as a σ1-targeting antiviral agent.

To validate the potential of catechin gallate and bilobetin as lead candidates against mammalian reovirus, 200 ns molecular dynamics (MD) simulations were conducted to study the stability and conformational landscape of the ligand-receptor system [86]. We compared both complexes with the apo protein, evaluating RMSD, RMSF, Rg, SASA and hydrogen bond. Analysis of the Cα root-mean-square deviation (RMSD) showed that the σ1-catechin gallate complex reached equilibrium rapidly with slight deviation and maintained a mean RMSD of $0.680 \pm 0.178$ nm. On the other hand, the apo protein showed prolonged volatility and a much higher average RMSD of 0.949 nm, which indicates that binding of the ligand (catechin gallate) succeeds in constraining the structural fluctuations. However, the σ1-bilobetin complex had a significantly more intense RMSD of $4.106 \pm 0.208$ nm, indicating a weaker ligand-receptor association. This stability of the complex was further analyzed by root mean square fluctuation (RMSF) analysis, which measures the local residue flexibility [87]. The σ1-catechin gallate complex exhibited a significantly smaller average RMSF value of $0.30 \pm 0.177$ nm than the dynamic apo structure ($0.62 \pm 0.297$ nm). Notably, residues in the active site pocket exhibited reduced mobility after ligand binding imputing that the binding pocket is rigidified by the ligand and maintains conformational integrity. Conversely, σ1-bilobetin complex maintained significant residue fluctuations with an average RMSF of $0.57 \pm 0.345$ nm, similar to apo protein. Assessment of the Radius of Gyration (Rg) confirmed that the association of ligand induces a compact protein architecture [88]. The complex was at a consistent, lower average Rg value of $1.89 \pm 0.20$ nm with negligible oscillation after 20 ns compared to the apo form, which was expanded and less stable with a mean value of $2.07 \pm 0.19$ nm. The same trend as the apo structure was found with the σ1-bilobetin complex which had a higher average Rg of $2.14 \pm 0.22$ nm and maintained fluctuations during the 200 ns simulation, indicating that bilobetin binding was not conducive to overall compactness. Regarding solvent exposure, the σ1-catechin gallate complex had a slightly higher solvent-accessible surface area (SASA) value of $114.44 \pm 1.4$ nm$^2$ than the σ1-bilobetin complex ($114.33 \pm 1.3$ nm$^2$), while the apo protein maintained the lowest average SASA value of $112.89 \pm 1.41$ nm$^2$. This slight increase in SASA value for both complexes is most likely an indication of local structural rearrangements to accommodate the ligand but not because of destabilization as the overall solvation profile was robust [89]. Finally, the thermodynamic stability of the interaction was emphasized by the persistence of intermolecular hydrogen bonds [90]. The existence of intermolecular hydrogen bonds throughout the 200 ns trajectory indicated that the ligand-σ1 complexes remained structurally stable and that these interactions were probably a major determinant of complex integrity.

The molecular dynamics trajectories were subjected to Principal Component Analysis (PCA) with the aim of attaining dimensionality reduction, and the extraction of the essential conformational variance [91]. The resulting data showed that the first three principal components (PC1, PC2, and PC3) explained about three-quarters of all dynamic motion in both the apo (74.9%) and the holo complexes (74.1% and 76.2% for catechin gallate and bilobetin, respectively). For the σ1-catechin gallate complex, the variance was split across PC1, PC2, PC3 with 41.33%, 22.01%, 10.73% respectively, whereas the corresponding values for the σ1-bilobetin complex were 47.74%, 17.54%, and 10.95%. Importantly, the first five eigenvectors adequately described the functional dynamics of both systems, which together accounted more than 81% of the prominent structural transitions. The projection of the trajectories on the PC1/PC2 subspace indicated a sharp contrast on the conformational sampling. The apo protein was found to have a broad and diffuse distribution of clusters, implying high intrinsic flexibility and exploration of a large ensemble of different configurations (Fig 5A). Conversely, the σ1-catechin gallate complex (Fig 5B) took up a much more localized and densely populated cluster, meaning that the motion was confined to a smaller part of the phase space. In comparison, the σ1-bilobetin complex had an equally broad and diffuse cluster distribution in the PC1/PC2 space, suggesting a comparatively flexible conformation similar to the apo protein, and significantly less constrained than the σ1-catechin gallate complex (Fig 5C). This constriction of the available conformational subspace is a strong indication that Catechin gallate binding stabilizes the reovirus

σ1 protein and strongly restricts the overall flexibility of the protein and imposes a more rigid and well- defined structural form.

The energetic basis of reovirus σ1 protein interaction with the ligands was quantitatively determined using the Molecular Mechanics-Generalized Born Surface Area (MM-GBSA) method, which is a well-investigated computational technique used to reliably estimate the theoretical binding free energy of small molecules with their biological targets [92]. Assessment of the σ1-catechin gallate complex revealed a very desirable mean binding free energy ($\Delta G_{bind}$) of −15.6097 ± 3.21 kcal/mol. In comparison, the σ1-bilobetin complex exhibited less desirable mean $\Delta G_{bind}$ of −13.7327 ± 5.44 kcal/mol as compared to the σ1-catechin gallate complex. The robust, invariably favorable binding free energy of the σ1-catechin gallate complex (Fig 6C) energetically fixes the target, which is directly supported by the PCA analysis indicating a highly constrained, dense conformational cluster in the PC1/PC2 space. The strong and stable energetic interactions successfully confine the complex into a well-defined conformation, reducing the available phase space drastically relative to the apo state. The σ1-bilobetin complex, on the other hand, has a weaker and highly oscillating energetic profile with transient neutral binding states (Fig 6D). This dynamical instability does not adequately restrain the binding pocket, which directly corresponds to the PCA results in which the σ1-bilobetin complex is characterized by a broad, high-entropy conformational landscape, which is a close resemblance to the flexible apo protein. Collectively, these orthogonal analyses support that the structural stabilization of the σ1 protein is due to the superior thermodynamic stability of catechin gallate.

In this study, the antiviral potential of catechin gallate and bilobetin was extensively explored using a variety of computational methods. Of the two compounds, catechin gallate showed consistently favorable results throughout the ADMET profiling, molecular docking, molecular dynamics (MD) simulations including RMSD and RMSF analysis, Rg, SASA, Hydrogen Bond analysis and PCA, as well as post-simulation MM-GBSA free energy calculations. Examination of trajectory snapshots of 0, 50, 100, 150, and 200 ns confirmed that catechin gallate remained stably accommodated within the active site pocket of σ1 protein over the course of 200 ns simulation (Fig 7). The ligand formed specific and stable interactions with important residues within the JAM-A binding domain, a region of σ1 protein that is essential for reovirus attachment to host cells [70]. These findings collectively point to catechin gallate as a promising bioactive compound obtained from plants with strong inhibitory potency against mammalian reovirus through the targeting and stabilization of the σ1 attachment protein and likely blocking σ1-mediated host cell entry. Although experimental validation by *in vitro* and *in vivo* assays is still necessary to ensure its therapeutic efficacy, such studies are usually limited by financial constraints, limited access to specialized instrumentation, and restricted biosafety requirements. The current *in silico* modelling offers useful mechanistic understanding to prioritize and guide further experimental studies to contribute to rational design and development of targeted antiviral therapeutics.

## Conclusion

The pathogenesis of the mammalian reoviruses is attributed to its outer capsid protein σ1, which mediates viral attachment and consequent host cell entry. Given the current lack of targeted therapeutics that can disrupt this entry mechanism, this study made use of a comprehensive *in silico* framework to point out bioactive phytochemicals as probable σ1 inhibitors. Through the rigorous process of ADMET profiling, molecular docking, molecular dynamics (MD) simulations, PCA and MM-GBSA calculations, catechin gallate (CID: 6419835) was identified as the potential candidate with high binding affiliate and stability. Even though these findings present a strong initial basis to be used in further research, they are highly computational and thus need to be supported by experiment and clinical proof before any therapeutic conclusions are made. The validation in physiologically relevant *in vitro* models, such as 3D organoids, should be followed by *in vivo* experiments in murine models to justify the antiviral potential of catechin gallate.

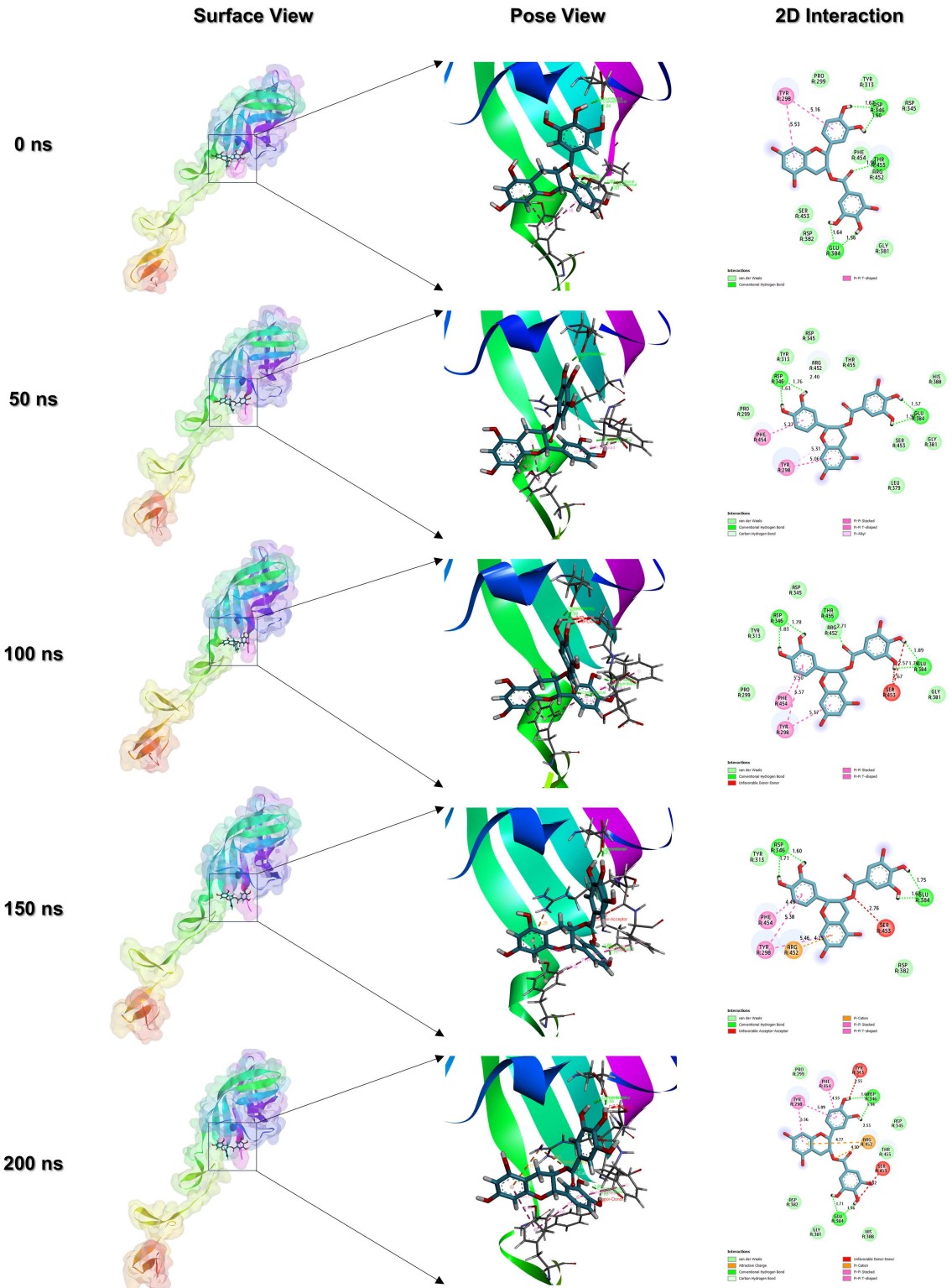

**Fig 7. Conformations of the σ1-catechin gallate complex during MD simulation.** Snapshots were taken at 0, 50, 100, 150, and 200 ns.

## Supporting information

**S1 File. Supporting information in ZIP format, including the list of phytochemicals, RMSD, RMSF, Rg, SASA, H-bond, MM-GBSA data, and all figures from the manuscript.**
(ZIP)

## Author contributions

**Conceptualization:** Eitu Dey, Shipon Dey, Leu Nandi, Md. Monirul Islam.

**Data curation:** Eitu Dey, Shipon Dey, Leu Nandi, Sayed Huzaifa Mumit, Refatul Arfat, Saifur Rahman Saif.

**Formal analysis:** Eitu Dey, Shipon Dey, Leu Nandi.

**Investigation:** Eitu Dey, Shipon Dey, Leu Nandi.

**Methodology:** Leu Nandi.

**Project administration:** Eitu Dey.

**Software:** Eitu Dey, Shipon Dey, Leu Nandi, Sayed Huzaifa Mumit.

**Supervision:** Md. Monirul Islam.

**Validation:** Eitu Dey, Shipon Dey, Leu Nandi, Md. Monirul Islam.

**Visualization:** Leu Nandi.

**Writing – original draft:** Eitu Dey, Shipon Dey, Leu Nandi, Sayed Huzaifa Mumit, Refatul Arfat, Saifur Rahman Saif.

**Writing – review & editing:** Leu Nandi, Sayed Huzaifa Mumit, Md. Monirul Islam.

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
