## [Decision Letter · Decision Letter 0]

26 Mar 2026

PONE-D-25-65098In-silico characterization of bioactive phytochemicals as antivirals targeting the Reovirus σ1 protein for inhibiting σ1-mediated host cell entryPLOS One

Dear Dr. Islam,

Thank you for submitting your manuscript to PLOS ONE. After careful consideration, we feel that it has merit but does not fully meet PLOS ONE’s publication criteria as it currently stands. Therefore, we invite you to submit a revised version of the manuscript that addresses the points raised during the review process.

Major modifications required as noticed by the reviewers. ==============================

We look forward to receiving your revised manuscript.

Kind regards,

Sajjad Ahmad

Academic Editor

PLOS One

Journal Requirements:

4. Please ensure that you refer to Figure 1 in your text as, if accepted, production will need this reference to link the reader to the figure.

5. We note that Figure(s) 1, 2, 3, 7 in your submission contain copyrighted images. All PLOS content is published under the Creative Commons Attribution License (CC BY 4.0), which means that the manuscript, images, and Supporting Information files will be freely available online, and any third party is permitted to access, download, copy, distribute, and use these materials in any way, even commercially, with proper attribution. For more information, see our copyright guidelines: http://journals.plos.org/plosone/s/licenses-and-copyright.

a. You may seek permission from the original copyright holder of Figure(s) 1, 2, 3, 7 to publish the content specifically under the CC BY 4.0 license.

Additional Editor Comments :

Major modifications required as noticed by the reviewers.

Reviewers' comments:

Reviewer's Responses to Questions

**Comments to the Author**

1. Is the manuscript technically sound, and do the data support the conclusions?

Reviewer #1: Partly

Reviewer #2: Yes

2. Has the statistical analysis been performed appropriately and rigorously? 

Reviewer #1: No

Reviewer #2: Yes

3. Have the authors made all data underlying the findings in their manuscript fully available?

Reviewer #1: Yes

Reviewer #2: Yes

4. Is the manuscript presented in an intelligible fashion and written in standard English?

Reviewer #1: No

Reviewer #2: Yes

5. Review Comments to the Author

Reviewer #1: 1. Limited Novelty

Catechin gallate is a well-characterized flavonoid with well-proven antiviral and bioactive effects. The manuscript does not adequately illustrate how the present findings appreciably improve current understanding beyond verifying its general antiviral potential through docking experiments. The authors could properly contextualize the novelty of targeting σ1 specifically and highlight what new mechanistic insights are gained.

2. Inadequate Docking Verification

There is no validation for the docking protocol. The following is not reported in the manuscript:

• If a co-crystallized ligand is available, redocking it

• The application of a reference inhibitor

• Comparing to established σ1 binders.

The reliability of docking scores (−8.1 kcal/mol) is questionable in the absence of validation. The authors ought to offer comparative benchmarking or protocol validation.

3. Overinterpretation of Computational Findings

A number of claims suggest that viral attachment and translational potential are functionally inhibited. Claims indicating steric blockage of host-cell entry or therapeutic promise should be limited because this study is purely computational. It should be made very clear in the conclusions that these results are predictive and need to be confirmed by experiments.

4. MM-GBSA and Sampling Concerns

Only 20 snapshots were used in the MM-GBSA analysis, which might not offer strong enough statistical confidence for a 200 ns trajectory. The authors ought to defend the choice of snapshots and think about boosting the sample density.

Minor Comments

1. Clarify whether the σ1 structure used corresponds to a full trimeric form during docking or a monomeric chain.

2. Provide more detail on grid box justification for docking.

3. Include standard deviations for RMSD/RMSF where appropriate.

4. Improve figure clarity (especially PCA plots and MM-GBSA bar chart labels).

5. Consider comparing catechin gallate with structurally related flavonoids to strengthen screening relevance.

Reviewer #2: Reviewer comments

The authors present a very interesting manuscript titled “In-silico characterization of bioactive phytochemicals as antivirals targeting the Reovirus σ1 protein for inhibiting σ1-mediated host cell entry”

” The presented study results are very interesting.

1 The manuscript is written clearly, and sections are well planned by the authors.

2 There are some minor changes regarding some typos and grammatical errors. Make sure that the citation is properly cited and add some updated citations.

3 Abstract should be rewritten with incorporation of proper computational results.

4 Introduction section is very short please make sure it has updated citations and some current investigation regarding the same topic portion must be revisited.

5 Figure resolutions are not in a proper readable format, make sure it is easy to understand.

6 Incorporate the pdb id in the methodology section as well with proper citation of the published articles.

7 Proper scientific format for species names is required throughout the journal.

8 Do comparative investigation with the binding energies along the PCA analysis

9 What residues were mostly involved in the binding energies?

6. PLOS authors have the option to publish the peer review history of their article (what does this mean?). If published, this will include your full peer review and any attached files.

Reviewer #1: No

Reviewer #2: No

---

## [Author Response · Author response to Decision Letter 1]

16 Apr 2026

Responses to Journal Requirements

1. Formatting and Style Requirements

We have revised the manuscript to ensure full compliance with PLOS ONE style and formatting guidelines, including appropriate file naming conventions.

2. Code Availability

No custom or author-generated code was used in this study. Therefore, the code sharing requirement is not applicable to our manuscript.

3. Data Availability

All relevant data supporting the findings of this study have been provided within the manuscript and its Supporting Information files. Specifically, the complete dataset has been included in the supplementary file (S1 File).

4. Figure Citation in Text

In the revised manuscript, Figure 1 has been properly cited at the beginning of the Materials and Methods section, as required.

5. Copyright Concerns for Figures

A separate detailed statement addressing the copyright concerns for Figures 1, 2, 3, and 7 has been prepared and submitted as an additional document titled:

“Response_to_Journal_Requirement_5_Copyright_Figures.docx”.

6. Citation of Additional References

Neither the reviewers nor the editor explicitly required the inclusion of additional references. Therefore, no new citations have been added based on this requirement.

Response to Reviewer 1

We sincerely thank the reviewer for the thorough evaluation of our manuscript and for the constructive and insightful comments. We have carefully considered all points and revised the manuscript accordingly where possible. Our detailed, point-by-point responses are provided below.

Major Comments

Comment 1:

Limited Novelty

Catechin gallate is a well-characterized flavonoid with well-proven antiviral and bioactive effects. The manuscript does not adequately illustrate how the present findings appreciably improve current understanding beyond verifying its general antiviral potential through docking experiments. The authors could properly contextualize the novelty of targeting σ1 specifically and highlight what new mechanistic insights are gained.

Response:

We thank the reviewer for this important and insightful comment. We agree that catechin gallate is a well-known bioactive flavonoid with reported antiviral properties. However, the focus of our study was not to claim catechin gallate as a newly discovered antiviral compound, but rather to evaluate, within a curated set of previously reported antiviral phytochemicals, which compound showed the most favorable interaction with the σ1 attachment protein of reovirus.

To clarify the novelty of the present work, our study emphasizes the target-specific computational evaluation of σ1, which is the key viral attachment protein responsible for mediating host-cell recognition and entry. By focusing on σ1, we attempted to identify a plausible mechanism by which catechin gallate may interfere with the σ1-JAM-A interaction, thereby potentially disrupting viral attachment. Thus, the novelty of the study lies in the comparative screening of reported antiviral phytochemicals against a biologically relevant reoviral attachment target, rather than in the discovery of catechin gallate itself.

In the manuscript, we aimed to conceptualize the rationale for targeting σ1 as the principal viral attachment protein involved in host-cell recognition and entry. Specifically, in lines 60-70 and 98-111, we outlined the biological significance of σ1 and its role in mediating interaction with junctional adhesion molecule-A (JAM-A). These sections were intended to justify the selection of σ1 as a relevant antiviral target and to explain how interference with the σ1-JAM-A interaction may potentially disrupt viral attachment. Thus, the manuscript frames the study as a target-specific computational evaluation focused on the early stage of reovirus infection.

Comment 2:

Inadequate Docking Verification

There is no validation for the docking protocol. The following is not reported in the manuscript:

• If a co-crystallized ligand is available, redocking it

• The application of a reference inhibitor

• Comparing to established σ1 binders.

The reliability of docking scores (−8.1 kcal/mol) is questionable in the absence of validation. The authors ought to offer comparative benchmarking or protocol validation.

Response:

We thank the reviewer for raising this important methodological concern. We fully agree that docking validation is desirable for strengthening confidence in the results. However, in the present case, the available σ1 protein structure used in this study does not contain a co-crystallized ligand, and after an extensive literature search, we were unable to identify a reported reference inhibitor or any established small-molecule σ1 binder that could be used for redocking or benchmarking.

Because of this limitation in the available structural and experimental data, we were not able to perform the specific validation procedures requested by the reviewer. To avoid overstating the findings, we have clarified in the revised manuscript, specifically in the conclusion, that the docking results should be interpreted as computational predictions rather than experimentally validated binding data.

We appreciate the reviewer’s point, and we agree that future studies should incorporate experimental validation once suitable reference ligands or validated σ1 binder data become available.

Comment 3:

Overinterpretation of Computational Findings

A number of claims suggest that viral attachment and translational potential are functionally inhibited. Claims indicating steric blockage of host-cell entry or therapeutic promise should be limited because this study is purely computational. It should be made very clear in the conclusions that these results are predictive and need to be confirmed by experiments.

Response:

We thank the reviewer for this crucial suggestion. We have carefully revised the manuscript to remove or soften statements that may have implied direct functional inhibition or therapeutic confirmation. In the revised text, we now present the findings as predictive computational evidence supporting a possible inhibitory interaction, rather than as proof of antiviral efficacy.

We have also revised the Conclusion to clearly state that the results are hypothesis-generating and require experimental validation before any biological or therapeutic inference can be made. This clarification has been added on page 16, lines 603–607.

Comment 4:

MM-GBSA and Sampling Concerns

Only 20 snapshots were used in the MM-GBSA analysis, which might not offer strong enough statistical confidence for a 200 ns trajectory. The authors ought to defend the choice of snapshots and think about boosting the sample density.

Response:

We thank the reviewer for this valuable recommendation. We agree that using only 20 snapshots may limit the robustness of the MM-GBSA estimate for a 200 ns simulation. In response, we have increased the sampling density substantially in the revised manuscript and recalculated the MM-GBSA binding energy using 200 snapshots. We believe this expanded sampling provides a more reliable estimate of the binding free energy and improves the statistical confidence of the analysis.

This revision has been incorporated on page 11, line 392, and is also reflected in Figure 5.

Minor Comments

Comment 1:

Clarify whether the σ1 structure used corresponds to a full trimeric form during docking or a monomeric chain.

Response:

We thank the reviewer for this important clarification request. In the docking study, we used one chain (chain A) of the σ1 trimer rather than the full trimeric assembly. This choice was made because the σ1 chains are identical, and using a single chain was sufficient for the receptor-binding analysis while also reducing computational complexity.

We have now clarified this point in the Materials and Methods section under “Extraction and purification of the protein structure,” (line 183) and the selected chain is also shown in Figure 2.

Comment 2:

Provide more detail on grid box justification for docking.

Response:

We thank the reviewer for this important suggestion. The grid box was defined to cover the receptor-binding region of σ1 that mediates interaction with junctional adhesion molecule-A (JAM-A) during viral attachment. Based on the key residues reported in cited Article 70, we positioned the grid box to include the full interface implicated in host recognition, ensuring that the docking analysis focused on the biologically relevant binding site.

This justification has now been added on page 6, lines 223–224.

Comment 3:

Include standard deviations for RMSD/RMSF where appropriate.

Response:

We thank the reviewer for this helpful suggestion. In the revised manuscript, the RMSD and RMSF results now include standard deviation values where appropriate.

Comment 4:

Improve figure clarity (especially PCA plots and MM-GBSA bar chart labels).

Response:

We thank the reviewer for this useful comment. The figures have been revised to improve clarity, and the PCA plot and MM-GBSA bar chart have been relabeled to remove ambiguity. In the revised manuscript, Figures 5 and 6 correspond to the PCA and MM-GBSA analyses, respectively.

Comment 5:

Consider comparing catechin gallate with structurally related flavonoids to strengthen screening relevance.

Response:

We sincerely thank the reviewer for this excellent suggestion. We have strengthened the comparative aspect of the study by including bilobetin, the second-ranked compound from our docking-based screening, alongside catechin gallate. The revised manuscript now compares both compounds across the relevant computational analyses, which helps to better contextualize why catechin gallate emerged as the most favorable candidate against the reovirus σ1 protein.

Response to Reviewer 2

We sincerely thank the reviewer for the careful evaluation of our manuscript and for the constructive suggestions. We have addressed all comments thoroughly, and the manuscript has been revised accordingly. Our point-by-point responses are provided below.

Comment 1:

The manuscript is written clearly, and sections are well planned by the authors.

Response:

We thank the reviewer for this positive and encouraging comment. We are pleased that the clarity of the writing and the organization of the manuscript were appreciated.

Comment 2:

There are some minor changes regarding some typos and grammatical errors. Make sure that the citation is properly cited and add some updated citations.

Response:

We thank the reviewer for this valuable suggestion. The manuscript has been carefully revised to correct typographical and grammatical errors throughout. All references and in-text citations have been thoroughly checked for accuracy and consistency. In addition, recent and relevant literature has been incorporated where appropriate. Specifically, citations 9, 10, 11, 12, 14, 15, 18, 19, 21, 22, 28, 44, 46, 51, and 81 have been updated.

Comment 3:

Abstract should be rewritten with incorporation of proper computational results.

Response:

We thank the reviewer for this important suggestion. The Abstract has been revised to incorporate the key computational findings presented in the Results section, improving its accuracy and informativeness. However, highly detailed molecular dynamics parameters (e.g., RMSD, RMSF, Rg, SASA, and hydrogen bond profiles) were not explicitly included, as their proper interpretation requires contextual explanation. Including such details in the Abstract may reduce clarity and readability. We believe the revised Abstract achieves an appropriate balance between completeness and conciseness.

Comment 4:

Introduction section is very short please make sure it has updated citations and some current investigation regarding the same topic portion must be revisited.

Response:

We thank the reviewer for this helpful suggestion. The Introduction has been revised and strengthened with updated and relevant references. Specifically, citations 9, 10, 11, 12, 14, 15, 18, 19, 21, 22, 28, 44, 46, and 51 have been updated. We also conducted an additional literature survey to identify recent advances related to this topic; however, no substantially new findings were identified that warranted major expansion beyond the revisions already made.

Comment 5:

Figure resolutions are not in a proper readable format; make sure it is easy to understand.

Response:

We thank the reviewer for this comment. All figures have been carefully revised and replaced with high-resolution versions to improve readability and visual clarity. The updated figures are now clearer and easier to interpret.

Comment 6:

Incorporate the pdb id in the methodology section as well with proper citation of the published articles.

Response:

We thank the reviewer for this important suggestion. The PDB ID of the protein used in this study has been incorporated into the Materials and Methods section under the subsection “Extraction and purification of the protein structure.” The corresponding published article has also been properly cited. This information is provided on page 5, line 179 (Reference 60).

Comment 7:

Proper scientific format for species names is required throughout the journal.

Response:

We thank the reviewer for pointing this out. The manuscript has been carefully revised to ensure that all species names are presented in the correct scientific format throughout.

Comment 8:

Do comparative investigation with the binding energies along the PCA analysis.

Response:

We thank the reviewer for this valuable suggestion. A comparative analysis integrating binding energy results with principal component analysis (PCA) has been added to better correlate the thermodynamic stability with the dynamic behavior of the complexes. This addition can be found on page 15, lines 564–574.

Comment 9:

What residues were mostly involved in the binding energies?

Response:

We thank the reviewer for this important clarification. The revised manuscript now includes information on the key residues contributing to binding energy. Additionally, graphical representations illustrating residue-wise energy decomposition have been included (Figures 5E and 5F). These details are provided on page 12, lines 414–415.

Response to Journal Requirement #5 – Regarding Copyright Concern on Figures 1, 2, 3, and 7

Dear Academic Editor and Editorial Board,

Thank you for bringing this to our attention. We would like to respectfully and clearly clarify the origin and preparation of Figures 1, 2, 3, and 7 in our manuscript.

All four figures are entirely original and were produced in two steps by our research team:

1. The core scientific visualizations including all protein-ligand interaction outputs were directly generated using Biovia Discovery Studio, a bioinformatics software platform that is freely available to the global research community for academic and scientific use.

2. The visual aesthetics of these figures such as labeling, layout, color adjustments, and presentation formatting were prepared using Microsoft PowerPoint, another widely available and freely accessible software tool.

At no point were these figures reproduced, adapted, or derived from any previously published work or third-party copyrighted material. They solely and exclusively represent the original computational results of our own study.

As both tools used (Biovia Discovery Studio and Microsoft PowerPoint) are freely accessible software platforms, and as all figures are original outputs of our own research workflow, we respectfully assert that no copyright infringement has occurred and that no third-party permission is required for publication under the CC BY 4.0 license.

As further evidence of our original work and editing process, we have included the original PowerPoint source files within the supplementary materials as S1 File, which contains all supporting information and clearly demonstrates that these figures were independently created and edited by our team.

We trust that this clarification, along with the supporting evidence provided in S1 File, will satisfactorily resolve the concern raised under Requirement #5. We remain happy to provide any further information if required.

Si

---

## [Decision Letter · Decision Letter 1]

8 May 2026

In silico characterization of bioactive phytochemicals as antivirals targeting the reovirus σ1 protein for inhibiting σ1-mediated host cell entry

PONE-D-25-65098R1

Dear Dr. Islam,

We’re pleased to inform you that your manuscript has been judged scientifically suitable for publication and will be formally accepted for publication once it meets all outstanding technical requirements.

Kind regards,

Sajjad Ahmad

Academic Editor

PLOS One

Additional Editor Comments (optional):

Accept

Reviewers' comments:

Reviewer's Responses to Questions

**Comments to the Author**

1. If the authors have adequately addressed your comments raised in a previous round of review and you feel that this manuscript is now acceptable for publication, you may indicate that here to bypass the “Comments to the Author” section, enter your conflict of interest statement in the “Confidential to Editor” section, and submit your "Accept" recommendation.

Reviewer #1: All comments have been addressed

Reviewer #2: (No Response)

2. Is the manuscript technically sound, and do the data support the conclusions?

Reviewer #1: Yes

Reviewer #2: Yes

3. Has the statistical analysis been performed appropriately and rigorously? 

Reviewer #1: N/A

Reviewer #2: Yes

4. Have the authors made all data underlying the findings in their manuscript fully available?

Reviewer #1: Yes

Reviewer #2: Yes

5. Is the manuscript presented in an intelligible fashion and written in standard English?

Reviewer #1: Yes

Reviewer #2: Yes

6. Review Comments to the Author

Reviewer #1: The manuscript entitled “In silico characterization of bioactive phytochemicals as antivirals targeting the reovirus σ1 protein for inhibiting σ1-mediated host cell entry” presents a comprehensive computational investigation of phytochemical inhibitors against the reovirus σ1 attachment protein. The study integrates molecular docking, ADMET profiling, molecular dynamics simulations, PCA, and MM-GBSA analyses to identify catechin gallate as a potential antiviral lead compound. The topic is scientifically relevant because of the emerging zoonotic significance of mammalian orthoreoviruses and the absence of approved therapeutics targeting viral entry mechanisms.

The manuscript is generally well organized and methodologically detailed. The revised version demonstrates noticeable improvement in response to reviewer concerns, particularly regarding MM-GBSA sampling, inclusion of standard deviations, clarification of docking methodology, and refinement of overly interpretative statements. The addition of comparative analyses between catechin gallate and bilobetin strengthens the computational framework.

Reviewer #2: (No Response)

7. PLOS authors have the option to publish the peer review history of their article (what does this mean?). If published, this will include your full peer review and any attached files.

Reviewer #1: No

Reviewer #2: **Yes:** Faisal Ahmad

---

## [Editor Report · Acceptance letter]

PONE-D-25-65098R1

PLOS One

Dear Dr. Islam,

I'm pleased to inform you that your manuscript has been deemed suitable for publication in PLOS One. Congratulations! Your manuscript is now being handed over to our production team.

Kind regards,

on behalf of

Dr. Sajjad Ahmad

Academic Editor

PLOS One